# MFCF-Gait: Small Silhouette-Sensitive Gait Recognition Algorithm Based on Multi-Scale Feature Cross-Fusion

**DOI:** 10.3390/s24175500

**Published:** 2024-08-24

**Authors:** Chenyang Song, Lijun Yun, Ruoyu Li

**Affiliations:** 1College of Information, Yunnan Normal University, Kunming 650500, China; 2223410012@ynnu.edu.cn (C.S.); 214100022@user.ynnu.edu.cn (R.L.); 2Engineering Research Center of Computer Vision and Intelligent Control Technology, Department of Education of Yunnan Province, Kunming 650500, China

**Keywords:** gait, gait recognition, deep learning, feature fusion, super-resolution

## Abstract

Gait recognition based on gait silhouette profiles is currently a major approach in the field of gait recognition. In previous studies, models typically used gait silhouette images sized at 64 × 64 pixels as input data. However, in practical applications, cases may arise where silhouette images are smaller than 64 × 64, leading to a loss in detail information and significantly affecting model accuracy. To address these challenges, we propose a gait recognition system named Multi-scale Feature Cross-Fusion Gait (MFCF-Gait). At the input stage of the model, we employ super-resolution algorithms to preprocess the data. During this process, we observed that different super-resolution algorithms applied to larger silhouette images also affect training outcomes. Improved super-resolution algorithms contribute to enhancing model performance. In terms of model architecture, we introduce a multi-scale feature cross-fusion network model. By integrating low-level feature information from higher-resolution images with high-level feature information from lower-resolution images, the model emphasizes smaller-scale details, thereby improving recognition accuracy for smaller silhouette images. The experimental results on the CASIA-B dataset demonstrate significant improvements. On 64 × 64 silhouette images, the accuracies for NM, BG, and CL states reached 96.49%, 91.42%, and 78.24%, respectively. On 32 × 32 silhouette images, the accuracies were 94.23%, 87.68%, and 71.57%, respectively, showing notable enhancements.

## 1. Introduction

Gait recognition refers to the technique of identifying individuals or analyzing behaviors by analyzing their walking or movement patterns. It utilizes unique motion patterns and biological characteristics generated by the human body during walking, converting them into identifiable features [1]. Due to its non-invasive and difficult-to-disguise nature, gait recognition holds significant application potential in forensic and security domains. However, it faces numerous challenges in practical applications due to the complexity of real-world scenarios.

From a data perspective, current research in gait recognition can be categorized into model-based and silhouette-based approaches [2]. Model-based methods construct models using relationships such as the positions of human body keypoints, utilizing 2D, 3D, or SMPL models to extract gait features through data learning [3]. Silhouette-based methods focus on extracting gait features from gait silhouette images. With the rise in deep learning, research in gait recognition has also integrated deep learning techniques. Among model-based methods, PoseGait [4] utilizes 3D body pose modeling, GaitGraph [5] employs graph convolutional networks on 2D skeletal models, and HMRGait [6] achieves end-to-end gait recognition using pre-trained human body mesh models based on the SMPL model [7]. Model-based methods exhibit robustness against noise from different clothing but suffer from accuracy issues when dealing with low-resolution footage or distant subjects, which significantly impacts recognition accuracy. In contrast, silhouette-based methods are widely applicable, are simple to implement, and require fewer computational resources. GaitSet [8] treats sequences of gait silhouettes as sets, using convolutional neural networks to extract gait sequence features and compressing temporal information through max pooling to approximate mathematical set features. This approach is influential due to its simplicity and efficiency. GaitPart [9] focuses on local details of gait silhouettes, combining spatial segmentation and temporal micro-motion capture modules to enhance feature extraction capabilities. GaitGL [10] combines global and local convolution operations to complement each other, addressing the drawbacks of purely global or local approaches. The GaitBase model from the OpenGait [11] project has been meticulously designed through extensive experimentation, resulting in the selection of several streamlined, widely used, and validated modules. These modules are integrated to form a concise yet robust baseline model.

In studies on silhouette-based gait recognition, it has been a long-standing practice to preprocess gait silhouettes using the Takemura method [12] to a size of 64 × 64 for training and testing purposes. Some models also utilize larger 128 × 128 silhouettes to achieve better recognition rates [13]. However, this approach overlooks the possibility that models in real-world applications may encounter smaller silhouettes, such as those with a resolution of 32 × 32. Compared to the larger sizes, 32 × 32 silhouettes lose more image details and information, leading to a decrease in recognition accuracy. Current mainstream silhouette-based gait recognition models like GaitSet, GaitPart, and GaitGL experience significant drops in accuracy when confronted with smaller silhouettes. In complex real-world scenarios, such as when dealing with human subjects at greater distances, original silhouette resolutions lower than 64 × 64 become common. This situation occurs, for instance, with standard high-definition surveillance cameras commonly used today, especially at distances exceeding 30 m. The significance of recognizing silhouettes from further distances lies in the potential to maintain high accuracy rates even with these smaller resolutions. This capability could substantially increase the detection range of cameras, thereby achieving cost reduction goals and other practical applications, making it highly valuable in real-world settings.

This paper proposes two methods to address issues caused by small silhouettes. First, a super-resolution interpolation algorithm is applied at the input stage to standardize input silhouette resolutions to 64 × 64, attempting to compensate for geometric detail losses in small silhouettes while normalizing data. The experimental results show improved accuracy across different resolution silhouettes. Additionally, enhancing the super-resolution interpolation algorithm during the Takemura method preprocessing stage benefits recognition rates for 64 × 64 silhouettes. Historically, there have been studies improving gait recognition algorithm accuracy through super-resolution before the popularity of deep learning, such as Makihara’s work [14] using multiple low-frame-rate gait sequences to construct high-frame-rate sequences based on temporal cycles and Zhang Jun’s approach [15] capturing lost high-frequency information combined with neighborhood embedding and interpolation methods. However, these were tailored for Gait Energy Images (GEI) [16] and did not integrate deep learning, potentially paving the way for future deep learning-based gait recognition super-resolution algorithms.

Furthermore, this paper introduces a multi-scale feature cross-fusion module in the model, allowing for the main network to continuously operate while preserving and merging low-resolution features containing more semantic information with high-resolution features at multiple stages. This compensates for image detail losses after multiple convolution and pooling operations at lower levels and sensitizes the model to small-scale silhouette information. Experimental results on the CASIA-B dataset demonstrate that improvements in input and model dimensions enhance recognition accuracy, surpassing models like GaitSet and GaitPart for 32 × 32 gait silhouettes, highlighting unique advantages. This study contributes to advancing gait recognition technology, particularly in handling challenges posed by smaller resolution gait silhouettes in practical applications.

## 2. Related Works

Our work primarily focuses on two aspects: dataset preprocessing and model design in silhouette-based gait recognition. Currently, the mainstream method for preprocessing in silhouette-based gait recognition is the Takemura method. The purpose of this method is to segment the original gait silhouette images into appropriate sizes while attempting to remove invalid information and unqualified images as much as possible. These processed images are then used for training or testing purposes. In addition to the Takemura method, our approach includes an additional step during preprocessing using super-resolution interpolation algorithms to standardize the input data to the same resolution. This process aims to enhance the detail information of smaller silhouette images. Regarding model design, we employ set pooling as the primary feature aggregation method. This approach has been validated for its effectiveness in numerous studies. The translation maintains the technical detail and clarity of the original text, ensuring that the key concepts related to dataset preprocessing and model design in silhouette-based gait recognition are accurately conveyed.

### 2.1. Takemura Method

Takemura method, introduced in 2018 by Noriko Takemura in collaboration with the OU-ISIR MVLP dataset, refers to a method for normalizing gait silhouette images [12]. Specifically, the method starts by identifying the top and bottom of the silhouette image to remove excess background. It then determines the horizontal center by statistically analyzing pixel values horizontally and adjusts the image height to 64 pixels while proportionally adjusting the width. Finally, adhering to symmetry principles, the silhouette image is cropped into a square shape. The normalization process concludes by resizing the silhouette image to either 64 × 64 or 128 × 128 resolution, making it suitable for deep learning applications. Figure 1 describes the process. During this normalization process, inevitably, the use of super-resolution algorithms is involved. However, the original Takemura method paper does not specify which super-resolution algorithm was used. This lack of specification may have led to inconsistent data processing across different models, resulting in varying results when the same model is reproduced in different studies. In this paper’s experimental section, we will compare and discuss the Takemura method’s performance when using different super-resolution techniques. The continuation provides a detailed explanation of the Takemura method’s steps and addresses the potential variability introduced by unspecified super-resolution algorithms in different research contexts.

Furthermore, when using the Takemura method for processing gait silhouette images in gait recognition, images with smaller silhouettes are often filtered out for various reasons. Figure 2 presents examples of potential elements that may be removed during the preprocessing stage. Specifically, during the filtering process, the cumulative value of the entire image is computed. Given that gait datasets are typically binary images, this cumulative value effectively reflects the size of the silhouette within the image. Images with a cumulative value below a certain threshold are removed, and the remaining images are then processed by the Takemura method to be resized to 64 × 64 pixels. Although the Takemura method inherently loses details during scaling, in practical applications, there is a possibility that the distance between the person and the camera may be too great, resulting in silhouette images significantly smaller than 64 × 64 pixels. This further loss of detail can severely impact the accuracy of the model.

### 2.2. Super-Resolution

For deep learning models, input data are crucial. On one hand, the dataset must comprehensively reflect the problem to facilitate effective model training. On the other hand, strict adherence to data formatting requirements is essential. For silhouette-based gait recognition models, different resolutions of gait silhouettes must be standardized to the same size for input into the model. Failure to match the expected data format with that of the model can lead to training failures or errors [17]. Super-resolution algorithms effectively address this issue. Another advantage of using this method to standardize gait silhouette sizes is that it eliminates the need for modifying the model when faced with silhouettes of varying sizes. Consequently, the model only needs to be trained once to handle gait silhouettes of all sizes, thereby resulting in a robust model.

Super-resolution algorithms are image processing techniques aimed at recovering high-resolution details from low-resolution images or videos. Their primary objective is to infer missing high-frequency details from limited data by leveraging inherent image information, structure, statistical regularities, and possible prior knowledge to enhance image quality. However, mathematically, super-resolution algorithms deal with ill-posed problems. The irreversible loss of high-frequency detail in original low-resolution images or videos means any algorithm can only attempt to compensate for this loss, leading to uncertainties in restoring high-resolution images. It is impossible to completely and accurately reconstruct the original high-resolution image from low-resolution data. Consequently, the performance of different super-resolution algorithms varies [18].

This study observes that the reproducibility of results for the same model often varies across many gait recognition-related papers [11]. Upon a comparative analysis, it was found that besides factors such as training environment and adjustments to hyperparameters, the choice of super-resolution algorithm significantly influences model training outcomes. In certain deep learning models, simply substituting a superior super-resolution algorithm has shown to improve accuracy compared to the original study. This phenomenon is attributed to the heightened importance of geometric details supplemented by super-resolution when silhouette image resolutions are notably low.

The most common types of super-resolution algorithms are various interpolation methods, such as nearest neighbor interpolation [19], bilinear interpolation [20], and bicubic interpolation [21]. Interpolation algorithms, compared to other types of super-resolution methods, offer greater flexibility and can be applied at each stage of gait recognition data processing, effectively enhancing model performance. Consequently, this paper primarily discusses the impact of various interpolation-based super-resolution algorithms on gait recognition.

Nearest neighbor interpolation is a relatively straightforward super-resolution algorithm that operates by duplicating each pixel from a low-resolution image to its corresponding position in a high-resolution image using the value of the nearest neighbor pixel for filling. This method is simple and intuitive with fast computation speed, but it lacks numerical smoothing and may result in images with jagged edges. The formula is as follows:(1)$$\begin{array}{c} G\left(i+u,j+v\right)=G\left(i,j\right) \end{array}$$

In this context, where u, v are decimal fractions within the range [0, 1), $G\left(i,j\right)$ represents the value at point $\left(i,j\right)$ in the low-resolution image.

Bilinear interpolation considers the weighted relationships between the adjacent four pixels, in contrast to nearest neighbor interpolation. Consequently, it often produces smoother and more accurate results. During the process of image enlargement, bilinear interpolation typically yields smoother and more natural outcomes. The formula is as follows:(2)$$\begin{array}{c} G\left(i+u,j+v\right)=\left(1-u\right)\left(1-v\right)G\left(i,j\right)+\left(1-u\right)vG\left(i,j+1\right)+\\ u\left(1-v\right)G\left(i+1,j\right)+uvG\left(i+1,j+1\right) \end{array}$$
where u, v are decimal numbers in the interval [0, 1) and $G\left(i,j\right)$ denotes the value at the low-resolution image point $\left(i,j\right)$.

Bicubic interpolation, which we ultimately adopted, represents a relatively advanced interpolation method compared to bilinear interpolation. It constitutes a further improvement over bilinear interpolation by involving more neighboring pixels in the interpolation calculation, thereby enhancing the quality and detail enhancement capability of the results. This method is capable of producing smoother and clearer enlarged images. It employs a cubic polynomial in the interpolation formula for numerical transformation, defined as follows:(3)$$\begin{array}{c} f\left(x\right)=\left\{\begin{array}{cc} {\left|w\right|}^{3}-{\left|w\right|}^{2}+1, & 0\leq \left|w\right|\leq 1\\ -{\left|w\right|}^{3}+5{\left|w\right|}^{2}-8\left|w\right|+4, & 1\leq \left|w\right|\leq 2\\ 0, & \left|w\right|>2 \end{array}\right. \end{array}$$

The interpolation formula is as follows:(4)$$G\left(i+u,\;j+v\right)=A\times B\times C$$
where the meanings of A, B, and C are as follows:(5)$$A=\left[\begin{array}{cccc} S(1+u) & S(u) & S(1-u) & S(2-u) \end{array}\right]$$
(6)$$B=\left[\begin{array}{cccc} G(i-1,j-1) & G(i-1,j) & G(i-1,j+1) & G(i-1,j+2)\\ G(i,j-1) & G(i,j) & G(i,j+1) & G(i,j+2)\\ G(i+1,j-1) & G(i+1,j) & G(i+1,j+1) & G(i+1,j+2)\\ G(i+2,j-1) & G(i+2,j) & G(i+2,j+1) & G(i+2,j+2) \end{array}\right]$$
(7)$$C=\left[\begin{array}{c} S(1+v)\\ S(v)\\ S(1-v)\\ S(2-v) \end{array}\right]$$

### 2.3. Set Pooling

The concept of set pooling was first introduced in the GaitSet, a classic model for gait recognition. This method takes into account that the number of gait silhouette images for a person can vary arbitrarily, sometimes resulting in significantly different lengths of gait sequences. Therefore, using the maximum pooling function along the temporal dimension to extract gait features can capture more generalized information. In recent years, this approach has been validated for its effectiveness across various studies. The rationale for choosing set pooling lies in several unique advantages: It treats gait sequences as sets, thus eliminating the strict temporal order requirement and reducing computational burden on hardware. Furthermore, with the increasing richness of gait recognition datasets, particularly in outdoor environments, set pooling has exceeded expectations, demonstrating robustness against visual noise and varying lengths of gait sequences [22]. It has outperformed many subsequent methods in complex real-world scenarios. The specific formula is as follows:(8)$$SP\left(Feature\left(n,c,h,w\right)\right)=Max(Feature\left(n,c,h,w\right),0)$$
where “Feature” denotes the feature matrix and n,c,h,w, respectively, represent the number of feature maps, channels, height, and width of the feature maps. “$Max(Feature\left(n,c,h,w\right),0)$” signifies aggregating the feature matrix along the dimension of feature map number n by retaining the maximum values, resulting in the aggregated feature as “$Feature\left(1,c,h,w\right)$”.

## 3. Materials and Methods

### 3.1. Dataset Detail

#### 3.1.1. Dataset

The CASIA-B dataset [23] is one of the most widely used datasets for gait analysis. It includes RGB and silhouette multi-view gait data from 124 participants, captured at 11 different angles ranging from 0° to 180° with an interval of 18°. This dataset incorporates three walking conditions: normal (NM), in a coat (CL), with a bag (BG). For each participant at each viewpoint, there are 6 sequences of normal walking gaits, 2 sequences of walking with a coat, and 2 sequences of walking with a bag, resulting in a total of 110 gait sequences per person across the 11 viewpoints.

#### 3.1.2. Dataset Splitting and Rank-1

The most commonly used testing protocol for the CASIA-B dataset is the subject-independent protocol, where individuals in the training set do not overlap with those in the testing set. A typical approach involves assigning data from the first 74 participants to the training set, leaving the data from the remaining 50 participants for testing. Within the testing set, gait sequences are categorized into probe set and gallery set [24].

In gait recognition systems, “Probe” typically refers to the probe set (or verification set), while “Gallery” can be understood as the gallery set (or enrollment set). In this system, for each known individual, there existed a corresponding set of gait sequences, which were combined into a gallery set. In the trained gait model, all sequences from the gallery set were inputted, resulting in the generation and storage of vector representations. During identification of a new gait sequence for verification, this sequence was inputted into the trained gait model, producing a new vector representation. The system calculated the Euclidean distance between the outputted vector and all vectors in the gallery set, subsequently computing the Rank-1 identification rate. If the new sequence matched the closest known individual in terms of Euclidean distance, the identification was considered successful; otherwise, it was deemed a failure. The Rank-1 identification rate was computed to assess the performance of the gait model. The computational details are illustrated in Figure 3.

The reason for adopting such a testing protocol was because it is impractical in real-world applications to pre-train the identities of individuals to be recognized. This approach allowed researchers to better simulate scenarios where gait recognition technology is applied in the real world, thus more reasonably assessing the performance of gait recognition algorithms on untrained individuals.

### 3.2. Methods

#### 3.2.1. Model Overview

As shown in Figure 4, the model in this paper was primarily divided into four parts: super-resolution processing, backbone network, multi-scale feature cross-fusion module (MFCF), and single-scale horizontal pyramid mapping (SHPM).

The role of super-resolution processing is twofold: First, it aims to compensate for lost image details in small silhouette images to enhance recognition rates [15]. Second, deep learning imposes strict requirements on input data formatting to avoid computational errors, necessitating uniform formatting. Furthermore, our experiments found that selecting a superior super-resolution algorithm also improved recognition rates for 64 × 64 silhouette images. The backbone network was constructed using the classic VGGNet [25], employing alternating convolutional and pooling layers to extract gait features, followed by set pooling (SP) to aggregate these features. The MFCF module integrated shallow and deep features from the backbone network. Shallow features contain rich geometric details but less semantic information, whereas deep features possess more semantic information but fewer geometric details. By collecting and fusing information from different layers, this module compensated for each other’s deficiencies [26]. Compared to other multi-scale feature fusion methods, MFCF ensures sufficient information exchange through iterative fusion of features across scales, minimizing information loss caused by pooling layers and making the model more sensitive to small silhouette images.

The SHPM includes the crucial component SHPP (Single-scale Horizontal Pyramid Pooling), adapted from Horizontal Pyramid Pooling (HPP) [27], initially used in pedestrian re-identification and later adopted in gait recognition. HPP achieves this by segmenting the feature maps from each module into strip-like regions at four different scales, thereby encouraging the neural network to focus on features of various sizes. The three-dimensional strip-like feature maps are then compressed into one-dimensional features through global max pooling and global average pooling, followed by summation. Finally, the pooled features are mapped and classified via a fully connected layer. HPP divides feature maps from various modules into strips at four different scales to encourage the neural network to focus on features of different sizes. It then uses global max pooling and global average pooling to compress the three-dimensional strip features into one-dimensional features, which are summed. Finally, the pooled features are mapped and classified through fully connected layers. However, our study found that the multi-scale segmentation method employed by HPP did not significantly benefit our model. Based on experimental results, a single scale proved effective. Therefore, we simplified HPP into the single-scale SHPP, reducing model parameters while preserving functionality. The specific structure of SHPP is shown in Figure 5.

#### 3.2.2. Multi-Scale Feature Cross-Fusion Module

In convolutional neural networks (CNNs), the utilization of pooling layers is a common operation known to significantly impact various aspects of model performance, such as feature extraction, enlarging receptive fields, enhancing translational invariance, and reducing computational complexity [28]. However, pooling layers also introduce irreversible loss of geometric details in feature maps, ultimately leading to high-level feature maps closer to discriminative modules lacking geometric details, thereby affecting the accuracy of smaller targets. Consequently, a method to address this issue involves fusing lower-level feature maps rich in geometric details with higher-level feature maps. Traditional multi-scale feature fusion methods either incorporate pooling layers within fusion modules, resulting in loss of geometric details, or fail to sufficiently integrate information across different scales, neglecting further exploration of fused features, thereby hindering the model’s ability to learn from small contour images [29].

In consideration of these challenges, this paper introduced a multi-scale feature cross-fusion (MFCF) module. The two branches of this module extract feature maps of specific sizes from different layers of the model. After a series of processing steps, these feature maps were combined with the output of the backbone network and fed into the SHPM (Scale-aware Hierarchical Pooling Module). The key characteristic of these specific layer feature maps was that after pooling, they resembled lower-resolution images that had lost some geometric details. Overall, after branch-wise inference and fusion, these feature maps were combined with the backbone network results and entered the discrimination stage. This process enhanced the weight of low-resolution image features in the final results, thereby guiding the model to focus on learning finer details of small silhouette features and ultimately improving the recognition accuracy of small-sized silhouette images. This module extracted feature maps from different levels of the model without internal pooling operations to preserve original resolutions and continued feature computation using convolutional kernels to augment semantic information of fused features [30]. Inter-branch operations involved cross-fusion through multiple stages with various convolutional kernels, employing techniques such as upsampling and downsampling to ensure comprehensive information exchange. Applying this approach to high-resolution image features yielded additional semantic information relevant to gait, which facilitated more accurate discrimination during the classification phase. Conversely, for low-resolution image features, this method recovered more of the lost geometric details, thereby contributing to improved accuracy during the discrimination stage.

Consequently, MFCF retained and explored detailed information while minimizing computational overhead, transmitting operation results alongside backbone network outcomes to discriminative modules to enhance the significance of multi-scale feature maps, thereby improving recognition rates for low-resolution gait silhouettes.

#### 3.2.3. Loss

This paper employed joint training of the model using Triplet Loss [31] and Cross Entropy Loss [32]. Triplet Loss is a commonly used loss function in gait recognition models, particularly advantageous for learning fine-grained features, as evidenced by numerous studies in the field. However, Triplet Loss may suffer from imbalanced sample quantities between classes during training, where some classes have fewer samples than others. This imbalance can lead to insufficient attention to samples from certain classes during training, resulting in unstable model training and slow convergence. Combining Triplet Loss with other loss functions in joint training can constrain these issues and compensate for these drawbacks. The formulation of the joint loss function is as follows:(9)$$Loss=a\times {Loss}_{tri}+b\times {Loss}_{ce}$$

The coefficients a and b correspond to the respective weighting factors for the two loss functions.

The objective of Triplet Loss is to learn to map samples from the same class into a compact feature space while differentiating samples from different classes into separate regions. In Triplet Loss, each training sample consists of three samples: an anchor sample, a positive sample, and a negative sample. Here, the anchor sample and the positive sample belong to the same class, whereas the negative sample belongs to a different class. The goal of the loss function is to minimize the distance between the anchor and the positive sample while maximizing the distance between the anchor and the negative sample. By adjusting the feature representations of these samples, the model can effectively distinguish between different classes. The computation of Triplet Loss is defined as follows:(10)$${Loss}_{tri}=max(0,\;d(a,\;p)-d(a,\;n)+margin)$$
where a denotes the feature representation of the anchor sample, p represents the feature representation of a positive sample from the same class as the anchor, and n represents the feature representation of a negative sample from a different class than the anchor. The function d(x,y) denotes the Euclidean distance or cosine similarity between features x, y. margin is a hyperparameter used to control the margin boundary, ensuring that the distance between samples from the same class is at least marginally greater than the distance between samples from different classes.

The objective of Cross Entropy Loss is to minimize the discrepancy between the predicted values of the model and the actual labels, ultimately reducing the gap between predicted results and true class categories. The computation of Cross Entropy Loss is as follows:(11)$${Loss}_{ce}=-\frac{1}{m}\sum_{i=1}^{m}\sum_{j=1}^{n}p_{ij}\ln q_{ij}$$
where m represents the total number of samples, n denotes the total number of classes, $p_{ij}$ equals 1 if the sample belongs to the class and 0 otherwise, and $q_{ij}$ represents the predicted probability that the sample belongs to the class.

## 4. Experimental Results

### 4.1. Implementation Details

The experimental setup and hyperparameters are crucial in deep learning. The experimental setup determines the platform on which the model runs, while appropriate hyperparameter settings greatly assist in model convergence. The experimental setup of this study is shown in Table 1, and the experimental hyperparameters are listed in Table 2.

### 4.2. Super-Resolution Algorithm Comparison Experiments

In silhouette-based gait recognition models, it is necessary to standardize gait silhouette images of varying sizes to a uniform resolution using super-resolution algorithms before processing. The most common practice is to adjust the resolution to 64 × 64 pixels. This study investigates the impact of different super-resolution algorithms on model accuracy during training. For gait silhouette images with larger original sizes and finer details in the CASIA-B dataset, this study directly compresses them to a resolution of 64 × 64 pixels using various super-resolution algorithms.

In addressing the case where gait silhouette images are smaller than 64 × 64 in practical applications, this paper employs the simplest interpolation method, namely nearest neighbor interpolation, to compress gait silhouette images from the CASIA-B dataset to 32 × 32. This simulates the loss of image details due to excessive distance in real scenarios. For silhouette images smaller than 64 × 64, the experimental design involves using various super-resolution algorithms to upscale them to a 64 × 64 resolution in order to investigate the effectiveness of different super-resolution algorithms in recovering image details. During training, the models uniformly use gait silhouette images compressed directly to 64 × 64 resolution. During testing, trained models process the corresponding directly compressed data and compress then upscale the data separately, yielding respective accuracy results. The experimental results are shown in Table 3.

Based on the data analysis from the experimental results table, the following conclusions can be drawn: After compressing larger silhouette images to a resolution of 64 × 64, the highest accuracy is achieved using bicubic interpolation, followed by bilinear interpolation, with nearest neighbor interpolation yielding the lowest accuracy. Regarding silhouette images at 32 × 32 resolution, bicubic interpolation exhibited the best accuracy in this study. The reason bicubic interpolation performs best in model recognition, as suggested by this paper, lies in its ability to fill in more local details, enabling the model to comprehensively capture feature information crucial for recognition. This approach allows the model to more accurately differentiate and identify differences in various gait features. Additionally, silhouette images processed through bicubic interpolation exhibit smooth grayscale transitions, providing probabilistic information beyond simple binary black-and-white images. The gradient transitions between high brightness areas and key points of the human body represent a confidence distribution, allocating more weight to regions with higher likelihoods. This enhances the reliability and reference capability of gait recognition models. The gait silhouettes processed by different super-resolution algorithms are illustrated in Figure 6. This achievement offers valuable guidance and insights for future gait recognition research and practical applications. Further exploration and enhancement of super-resolution algorithms, coupled with the fusion of local details and probabilistic information, hold promise for improving the performance and accuracy of gait recognition.

### 4.3. Feature Fusion Module Ablation Experiments

To validate the effectiveness of the proposed multi-scale feature cross-fusion module, particularly in exploring the accuracy on 32 × 32 silhouette images, experiments were conducted by modifying the base model of this paper to exclude this module and instead integrate other multi-scale feature fusion modules for comparison. The experimental setup involved four approaches: utilizing a unified super-resolution algorithm without additional modules, integrating the MGB module [8] from the original GaitSet model, incorporating the FPN module, and employing the method proposed in this paper. The MGB module, derived from the GaitSet model, aims to collect and fuse features from different depths to enhance representation. FPN [33] (Feature Pyramid Network) is a classic feature extraction structure that performs feature fusion across different levels of feature maps through both top-down and bottom-up pathways, effectively extracting feature representations with rich semantics and multi-scale information. The experimental results are presented in the data table. The experimental data for the different modules are presented in Table 4.

Based on the data presented above, it is evident that our proposed method not only outperforms others in accuracy on 64 × 64 silhouette images but also significantly surpasses alternative modules in performance on 32 × 32 silhouette images, making it particularly suitable for recognizing small-scale silhouettes.

### 4.4. Comparison with the Existing Methods

In comparison with current mainstream silhouette-based gait recognition models, our model demonstrates superior performance, particularly when handling small-scale silhouettes. Under identical hardware configurations and utilizing a unified super-resolution algorithm, our model is compared with GaitSet, GaitPart, and GaitGL on 64 × 64 and 32 × 32 silhouette images. The experimental results are presented in Table 5.

Based on the tabulated experimental results, it is evident that the model proposed in this paper exhibits outstanding performance across different image resolutions. Specifically, on 64 × 64 silhouette images, our model performs exceptionally well, surpassing the GaitSet model and achieving accuracy comparable to that of the GaitPart model. However, the accuracy does not surpass that of the GaitGL and GaitBase models. This may be due to the fact that in the MFCF module, a larger number of low-resolution image features are processed and combined with the backbone network features before entering the subsequent module. While this approach increases the discrimination weight of low-resolution images within the branch, it also dilutes the weight of high-resolution image features processed by the backbone network. Additionally, the simplicity of the backbone network used in this study might not have effectively captured the information from high-resolution images, thereby reducing the accuracy of the 64 × 64 silhouette images. On 32 × 32 silhouette images, our model demonstrates significant superiority over these three models. These findings indicate that the model modifications and the choice of super-resolution algorithm in this study are highly effective. On one hand, by modifying and optimizing the model structure while retaining the advantages of the original model, this study successfully enhances the overall model performance. On the other hand, the study identifies and to some extent addresses the often overlooked issue of small-scale silhouette images in current mainstream research. This issue holds significant practical importance, as small-scale silhouettes are more prevalent in certain real-world scenarios.

## 5. Conclusions

This paper addresses the issue in practical applications of silhouette-based gait recognition, where the accuracy of models is affected by excessively small silhouette images. To tackle this challenge, this study thoroughly investigates the impact of super-resolution algorithms at the input stage on gait recognition accuracy. The research reveals that employing super-resolution algorithms capable of enhancing image details with smooth transitional properties significantly improves model accuracy, particularly noticeable in the recognition of small-scale silhouette images. Furthermore, the paper proposes a novel multi-scale fusion module in the model architecture. This module is designed to better leverage multi-scale information, focusing on learning from small-scale details, thereby further enhancing the accuracy and robustness of the model towards small-scale silhouette images. Through these investigations, the effectiveness of model modifications and super-resolution algorithms is validated, demonstrating their crucial role in enhancing model performance and robustness. This advancement contributes to bridging the gap between gait recognition research and practical applications.

## Figures and Tables

**Figure 1 sensors-24-05500-f001:**
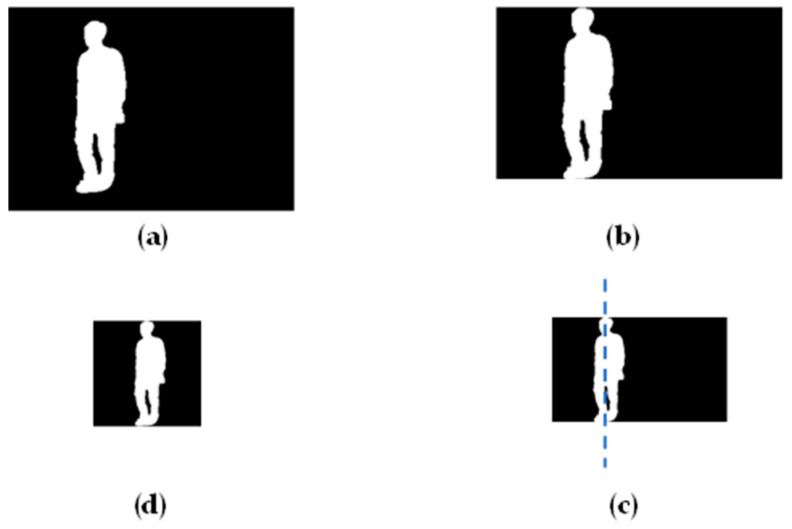
The Takemura method workflow: (**a**) begin with the original image, (**b**) trim excess background from the top and bottom, (**c**) resize height to 64 pixels and find the horizontal center of the contour image, and (**d**) crop the image width to 64 pixels.

**Figure 2 sensors-24-05500-f002:**
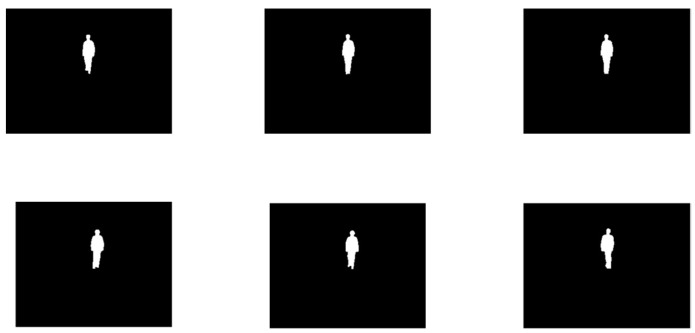
Some images removed during the preprocessing of the CASIA-B dataset.

**Figure 3 sensors-24-05500-f003:**
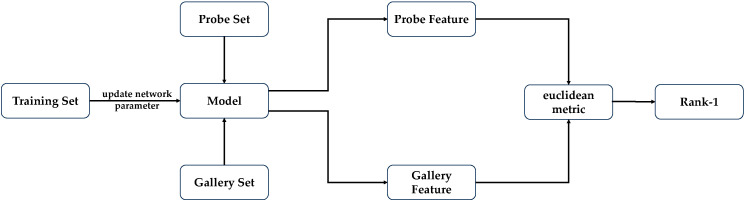
The method for calculating recognition rates in gait recognition algorithms.

**Figure 4 sensors-24-05500-f004:**
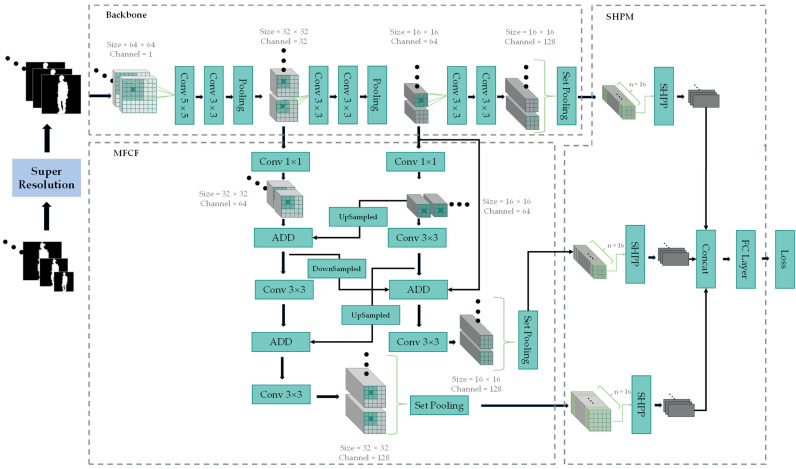
The overall architecture of the MFCF-Gait network.

**Figure 5 sensors-24-05500-f005:**
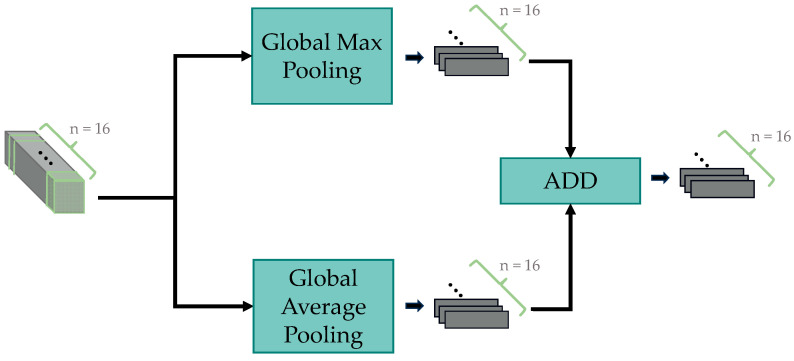
Detailed structure of SHPP.

**Figure 6 sensors-24-05500-f006:**
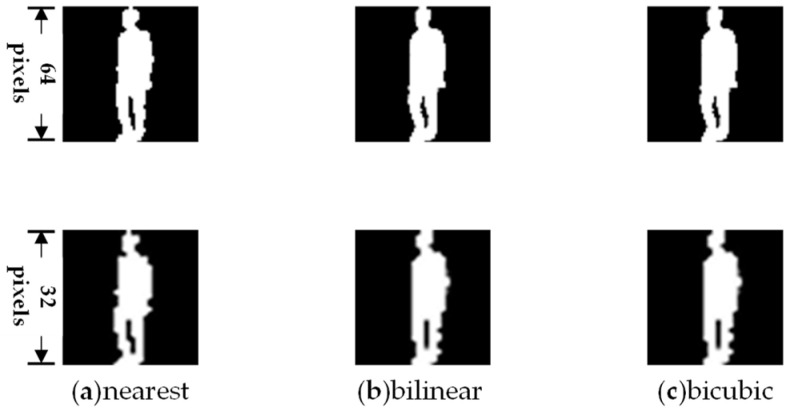
The gait silhouette images processed by different super-resolution algorithms. The first row shows images at a resolution of 64 × 64 pixels, while the second row displays images at a resolution of 32 × 32 pixels. (**a**) Nearest neighbor interpolation, (**b**) bilinear interpolation, (**c**) bicubic interpolation.

**Table 1 sensors-24-05500-t001:** Experimental configuration.

Environment Name	Configure Parameters
Operating system	Windows10
GPU	NVIDIA RTX 3060
CUDA	11.8
Compiled language	Python 3.9
Open source framework	Pytorch 1.12

**Table 2 sensors-24-05500-t002:** Training hyperparameters.

Hyperparameters	Value
iter	40,000
Optimizer	SGD
Learning Rate	0.1
Momentum	0.9
Weight Decay	0.0005
Learning Rate Schedule	MultiStep
Milestones	10,000 20,000 30,000
gamma	0.1

**Table 3 sensors-24-05500-t003:** Super-resolution algorithms comparison experiments.

Resolution	Nearest Neighbor	Bilinear	Bicubic	ACC/%		
				NM	BG	CL
64 × 64	✓			95.0	87.7	71.4
		✓		96.1	90.7	76.0
			✓	96.5	91.4	78.2
32 × 32	✓			86.5	75.4	53.7
		✓		87.0	77.8	57.1
			✓	94.2	87.7	71.6

**Table 4 sensors-24-05500-t004:** Feature fusion module ablation experiments.

Resolution	MGB	FPN	MFCF (Ours)	ACC/%		
				NM	BG	CL
64 × 64				95.9	90.2	76.8
	✓			96.4	91.6	75.5
		✓		96.0	90.4	76.4
			✓	96.5	91.4	78.2
32 × 32				83.8	75.8	54.6
	✓			85.9	77.5	54.9
		✓		90.3	82.6	63.4
			✓	94.2	87.7	71.6

**Table 5 sensors-24-05500-t005:** Model comparison experiments.

Resolution	Type	Gallery	0~180°											Mean
		Probe	0°	18°	36°	54°	72°	90°	108°	126°	144°	162°	180°	
64 × 64	NM	GaitSet	90.8	97.9	99.4	96.9	93.6	91.7	95.0	97.8	98.9	96.8	85.8	95.0
		GaitPart	94.1	98.6	99.3	98.5	94.0	92.3	95.9	98.4	99.2	97.8	90.4	96.2
		GaitGL	96.0	98.3	99.0	97.9	96.9	95.4	97.0	98.9	99.3	98.8	94.0	97.4
		GaitBase	95.6	99.2	100	99.0	97.6	95.4	97.5	99.4	100	99.1	94.2	97.9
		MFCF-Gait (Ours)	93.8	99.2	99.6	98.1	94.9	93.3	96.6	98.7	98.6	98.5	90.1	96.5
	BG	GaitSet	83.8	91.2	91.8	88.8	83.3	81.0	84.1	90.0	92.2	94.4	79.0	87.2
		GaitPart	89.1	94.8	96.7	95.1	88.3	94.9	89.0	93.5	96.1	93.8	85.8	91.5
		GaitGL	92.6	96.6	96.8	95.5	93.5	89.3	92.2	96.5	98.2	96.9	91.5	94.5
		GaitBase	92.5	95.7	96.1	95.7	92.1	90.2	92.2	95.3	97.1	95.4	89.9	93.8
		MFCF-Gait (Ours)	89.8	94.8	95.0	94.0	89.4	85.5	88.8	93.2	94.9	94.7	85.5	91.4
	CL	GaitSet	61.4	75.4	80.7	77.3	72.1	70.1	71.5	73.5	73.5	68.4	50.0	70.4
		GaitPart	70.7	85.5	86.9	83.3	77.1	72.5	76.9	82.2	83.8	80.2	66.5	78.7
		GaitGL	76.6	90.0	90.3	87.1	84.5	79.0	84.1	87.0	87.3	84.4	69.5	83.6
		GaitBase	68.0	79.9	82.7	81.5	77.4	75.5	76.8	79.9	80.6	78.7	67.5	77.1
		MFCF-Gait (Ours)	72.3	84.0	85.2	81.5	77.3	74.0	76.8	79.8	81.4	81.2	67.1	78.2
32 × 32	NM	GaitSet	64.4	79.6	88.1	88.9	80.0	75.8	81.1	86.3	86.8	79.8	60.7	79.2
		GaitPart	70.6	87.3	92.9	91.0	84.3	79.8	86.7	89.8	92.6	85.0	67.1	84.3
		GaitGL	73.5	90.6	93.2	91.5	86.9	83.9	86.8	91.6	94.3	90.5	67.9	86.4
		GaitBase	84.1	96.5	98.5	95.4	90.1	88.7	92.0	96.1	97.7	94.4	80.2	92.2
		MFCF-Gait (Ours)	86.5	95.8	98.9	97.6	94.0	91.8	95.4	98.1	97.5	95.3	85.5	94.2
	BG	GaitSet	55.7	70.2	76.4	79.9	69.4	64.2	70.0	81.3	77.6	74.8	51.9	70.1
		GaitPart	62.0	77.8	83.1	81.6	74.5	68.6	77.5	81.6	84.2	74.6	54.6	74.6
		GaitGL	68.1	86.0	88.5	86.2	81.5	77.3	80.2	86.6	88.6	86.9	60.3	80.9
		GaitBase	74.2	86.6	91.5	88.9	86.1	82.8	86.2	89.4	91.5	86.7	80.2	85.2
		MFCF-Gait (Ours)	80.1	89.8	93.5	93.1	88.1	83.0	87.0	92.8	92.0	87.9	77.2	87.7
	CL	GaitSet	35.0	48.4	58.5	57.4	55.3	50.9	55.9	56.9	52.8	46.8	32.8	50.0
		GaitPart	38.3	60.2	64.8	66.3	62.1	56.9	61.7	67.7	61.1	49.4	36.1	56.8
		GaitGL	45.0	71.8	77.5	72.6	69.9	63.1	64.1	69.3	67.8	62.3	39.4	63.9
		GaitBase	45.0	64.3	70.1	69.6	67.1	61.6	64.4	67.5	63.7	61.2	43.1	61.6
		MFCF-Gait (Ours)	58.7	73.9	79.8	75.7	74.3	70.6	73.2	76.7	76.6	72.8	55.0	71.6

## Data Availability

This study conducted experiments using the CASIA-B dataset. This dataset is provided by the Center for Biometrics and Security Research at the Chinese Academy of Sciences.

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
