# Peer review of "MFCF-Gait: Small Silhouette-Sensitive Gait Recognition Algorithm Based on Multi-Scale Feature Cross-Fusion"

_sensors, 2024, doi:10.3390/s24175500_

Round 1

Reviewer 1 Report

Comments and Suggestions for Authors

Manuscript: MFCF-Gait: Small silhouette-sensitive gait recognition algorithm based on multi-scale feature cross-fusion

This paper proposes a gait recognition system, Multi-scale Feature Cross Fusion Gait (MCFC-Gait), where accuracy of models may be impacted by smaller silhouette images.

Overall significance of the paper: 

Enhances the accuracy of gait models utilizing small-scale silhouette images.

In the following sections, group major and minor concerns and list major first.

Adequacy and accuracy of the literature review and theoretical rationale:

Literature review was thorough and well written.

Adequacy and accuracy of the methodology:

Some of the methodology was out of the realm of my expertise as a clinician, but based upon my review I thought it was thorough. 

Adequacy and accuracy of the presentation and statistical treatment of the results:

Some of the analysis was out of the realm of my expertise as a clinician, but based upon my review I thought it was thorough. 

The accuracy and clarity of figures and tables:

Figures and tables were clear and easy to interpret.

The accuracy and clarity of the discussion and interpretation:

Discussion was clear and easy to follow.  Some of the interpretation was out of the realm of my expertise as a clinical, but based upon my review, it was comprehensive.

Author Response

Dear reviewer,

Thank you very much for reading and reviewing the manuscript. We are honored that our work has received your recognition. 

Reviewer 2 Report

Comments and Suggestions for Authors

This article introduces a small contour-sensitive gait recognition algorithm named MFCF-Gait. The algorithm uses a super-resolution algorithm to preprocess the data at the input stage to enhance the image details. In addition, a multi-scale feature cross-fusion network model is introduced, which emphasizes the small-scale details by fusing the low-level feature information of high-resolution images with the high-level feature information of low-resolution images, so as to improve the recognition accuracy of small-contour images. Comments on the manuscript review are as follows:

1.       In practice, how to process contour images smaller than 64×64 pixels to avoid losing detail and significantly affecting the accuracy of the model?  

2.       According to the experimental results, why is the performance of the proposed method not the best at the original resolution of 64×64? Furthermore, if existing super-resolution modules were used, would incorporating them into the current method still yield a good performance?

3.       In the paper, you validated your method on only one dataset. Why didn't you also validate the effectiveness of your proposed method on other datasets like OUMVLP?

4.       The methods you compared with in the paper are from 2021 at the latest. Why didn't you try comparing with the latest state-of-the-art methods?

5.       The paper mentions a key component, SHPP (Single-scale Horizontal Pyramid Pooling), in the SHPM module. It is suggested to enhance its structure to make the entire framework more complete.

6.       What is the impact of different super-resolution algorithms on the accuracy of the model during training? Is there an algorithm that works better when working with larger profile images?

7.       How to improve the recognition accuracy of small-size contour images by fusing the low-level feature information of high-resolution images with the high-level feature information of low-resolution images?

Reviewer 3 Report

Comments and Suggestions for Authors

This paper proposes a new gait recognition system called MFCF-Gait. This system uses super-resolution algorithms and a multi-scale feature cross fusion network model to solve problems arising from small silhouette images. Results using the CASIA-B dataset showed that recognition accuracy was significantly improved compared to the baseline method, especially for small silhouette images.

The introduction of a super-resolution algorithm to process small silhouette images and the multi-scale feature cross fusion that integrates high-resolution and low-resolution features to improve the recognition accuracy of small silhouette images stand out. In addition, the reliability of the results was ensured through extensive experiments including various resolutions and conditions.

However, some additional explanation is needed as follows.

1. How does the model behave as the size of the silhouette image changes?

For example, it is necessary to explain how the model differs when a small image is input and when a large image is input.

2. How can the proposed method be applied in real scenarios?

3. Concrete examples of performance in real situations with the addition of various conditions

Round 2

Reviewer 2 Report

Comments and Suggestions for Authors

Authors have improved the manuscript in the revision considering the reviewers' comments. I support the publication of this manuscript with no further comments on its contents.